# Concurrent wet and dry hydrological extremes at the global scale

Paolo De Luca[1,2,3,4], Gabriele Messori[2,5], Robert L. Wilby[1], Maurizio Mazzoleni[2,3], and Giuliano Di Baldassarre[2,3]

[1]Geography and Environment, Loughborough University, Loughborough, LE11 3TU, United Kingdom

[2]Department of Earth Sciences, Uppsala University, Uppsala, 75236, Sweden

[3]Centre of Natural Hazards and Disaster Science (CNDS), Uppsala, 75236, Sweden

[4]Department of Water and Climate Risk, Vrije Universiteit Amsterdam, Amsterdam, 1081 HV, The Netherlands

[5]Department of Meteorology, Stockholm University and Bolin Centre for Climate Research, Stockholm, 10691, Sweden

**Correspondence:** Paolo De Luca, Department of Water and Climate Risk, Vrije Universiteit Amsterdam, Amsterdam, 1081 HV, The Netherlands, p.deluca@vu.nl

**Abstract.** Multi-hazard events can be associated with larger socio-economic impacts than single-hazard events. Understanding the spatio-temporal interactions that characterise the former is, therefore, of relevance to disaster risk reduction measures. Here, we consider two high-impact hazards, namely wet and dry hydrological extremes, and quantify their global co-occurrence. We define these using the monthly self-calibrated Palmer Drought Severity Index based on the Penman-Monteith model (sc_PDSI_pm), covering the period 1950-2014, at 2.5° horizontal resolution. We find that the land areas affected by extreme wet, dry and wet-dry events (i.e. geographically remote, yet temporally co-occurring wet or dry extremes) are all increasing with time, of which trends in dry and wet-dry episodes are significant (*p*-value <<0.01). The most geographically widespread wet-dry event was associated with the strong La Niña in 2010. This caused wet-dry anomalies across a land area of 21 million km2 with documented high-impact flooding and drought episodes spanning diverse regions. To further elucidate the interplay of wet and dry extremes at a grid-cell scale, we introduce two new metrics: the wet-dry (WD) ratio and the extreme transition (ET) time intervals. The WD-ratio measures the relative occurrence of wet or dry extremes, whereas ET quantifies the average separation time of hydrological extremes with opposite signs. The WD-ratio shows that the incidence of wet extremes dominates over dry extremes in the USA, northern and southern South America, northern Europe, north Africa, western China and most of Australia. Conversely, dry extremes are more prominent in most of the remaining regions. The median ET for wet to dry is ~27 months, while the dry to wet median ET is 21 months. We also evaluate correlations between wet-dry hydrological extremes and leading modes of climate variability, namely the: El Niño–Southern Oscillation (ENSO), Pacific Decadal Oscillation (PDO) and Atlantic Multi-decadal Oscillation (AMO). We find that ENSO and PDO have a similar influence globally, with the former significantly impacting (*p*-value <0.05) a larger area (18.1% of total sc_PDSI_pm area) compared to the latter (12.0%), whereas the AMO shows an almost inverse pattern, and significantly impacts the largest area

overall (18.9%). ENSO and PDO show most significant correlations over northern South America, central and western USA, the middle-East, eastern Russia and eastern Australia. On the other hand, the AMO shows significant associations over Mexico, Brazil, central Africa, the Arabic peninsula, China and eastern Russia. Our analysis brings new insights on hydrological multi-hazards that are of relevance to governments and organisations with globally distributed interests. Specifically, the multi-hazard maps may be used to evaluate worst-case disaster scenarios considering the potential co-occurrence of wet and dry hydrological extremes.

**Keywords:** multi-hazards; PDSI; wet-dry; metrics; hydrological extremes; compound extremes

## 1 Introduction

Natural hazards can interact in diverse ways, leading to multi-hazard events that can exacerbate disaster losses when compared to single-hazard occurrences (Zscheischler et al., 2018). Examples of multi-hazards are the co-occurrence of heavy precipitation or flooding with wind damage from extra-tropical cyclones (De Luca et al., 2020, 2017; Waliser and Guan, 2017), storm surge combined with fluvial flooding in deltas (Ward et al., 2018), flood episodes along with droughts (Collet et al., 2018) and landslides triggered by earthquakes (Kargel et al., 2016). Such combinations can lead to situations beyond the worst-case scenario planned by emergency managers, (re)insurance companies, businesses and governments and thus present a critical challenge for disaster risk reduction (Zscheischler et al., 2018). The relevance of multi-hazards has been recognised by scientific and stakeholder communities, and both have devoted significant efforts to the topic over the past decade (e.g. Forzieri et al., 2016; Gallina et al., 2016; Gill and Malamud, 2014; Kappes et al., 2012; Terzi et al., 2019; Zscheischler et al., 2018). Indeed, the United Nations Sendai Framework for Disaster Risk Reduction (UNISDR, 2015) now advocates multi-hazard approaches to disaster risk reduction.

Analysis of multi-hazards is highly relevant given anthropogenic climate change. Events such as floods and droughts already have significant humanitarian and socio-economic impacts (Alfieri et al., 2016; Barredo, 2007; Di Baldassarre et al., 2010; Jonkman, 2005; Naumann et al., 2015; Van Loon et al., 2016; Zhang et al., 2011), and are expected to become more frequent and/or severe in the future (Arnell and Gosling, 2016; Dai, 2012, 2011a; Hirabayashi et al., 2013; Hirsch and Archfield, 2015; IPCC, 2012; Milly et al., 2002), albeit with a large degree of uncertainty (e.g. Orlowsky and Seneviratne, 2013). Numerous studies have investigated the combination of flood and drought events or, more generally, wet and dry hydrological extremes at local and regional scales, for both present and future climates (e.g. Berton et al., 2017; Collet et al., 2018; Deangelis et al., 1984; Di Baldassarre et al., 2017; Gil-Guirado et al., 2016; Oni et al., 2016; Parry et al., 2013; Pechlivanidis et al., 2017; Quesada-Montano et al., 2018; Yan et al., 2013; Yoon et al., 2018). Examples include the analysis of abrupt drought-flood transitions in river basins in China (Yan et al., 2013) and in England and Wales (Parry et al., 2013). There is also the dynamical

interplay between society and hydrological extremes, intended as the mutual influence of human activities on floods and droughts (Di Baldassarre et al., 2017) and indices assessing the long-term evolution of vulnerability and adaptation to these hazards (Gil-Guirado et al., 2016). Other studies consider wet-dry interactions from a statistical perspective (Collet et al., 2018), or have related these two independent hazards to large-scale modes of climate variability (Cai and Rensch, 2012; Lee et al., 2018; Nobre et al., 2017; Siegert et al., 2001; Ward et al., 2014; Yoon et al., 2018).

Quantifying wet and dry (also extreme) hydrological events at both regional and global scales is a non-trivial task. Some commonly used metrics include the Palmer Drought Severity Index (PDSI) (Dai et al., 2004; Palmer, 1965), the Standardized Precipitation Index (SPI) (McKee T.B., Doesken N.J., 1995; McKee et al., 1993) and the Standardized Precipitation Evapotranspiration Index (SPEI) (Vicente-Serrano et al., 2010). For instance, the PDSI was used to evaluate the combined effect of the Pacific Decadal Oscillation (PDO) and El Niño Southern Oscillation (ENSO) on global wet and dry changes over land, showing that when these two modes are in phase (e.g. El Niño-warm PDO) wet and dry events are amplified (Wang et al., 2014). The PDSI and SPEI have also been used to quantify wet and dry trends over China, with generally good agreement between the two (Chen et al., 2017). At the global scale, the SPI and SPEI were used to explore wet and dry links with ENSO, PDO and the North Atlantic Oscillation (NAO) (Sun et al., 2016). The study found that ENSO has the greatest spatial impact for wet and dry changes, followed by the PDO having an effect in North America and eastern Russia, and the NAO affecting Europe as well as north Africa. The SPI has also been used in a global multi-model ensemble analysis of future projections in pluvial and drought events (Martin, 2018). This revealed that more severe pluvial events are expected in regions that are already wet and the same applies for more severe droughts in dry areas, although the overall "wet gets wetter, dry gets drier" paradigm may have some limitations, since when the paradigm is applied over land it does not hold as expected because of changes in atmospheric circulation, horizontal gradients of temperature and relative humidity (e.g. Byrne and O'Gorman, 2015; Yang et al., 2018).

In this study, we adopt a relatively broad definition of multi-hazard events, i.e. the temporal (yet spatially separate) co-occurrence of wet and dry hydrological extremes at the global scale, quantified following De Luca et al. (2017). We emphasize that the term "hydrological extreme" does not necessarily imply *observed* flooding or drought events, unless explicitly mentioned, and we always make use of this term when referring to the sequential occurrence of extremes with opposite sign (i.e. wet and dry). The relevance of both types of multi-hazards is evident. Stakeholders with geographically diverse portfolios, such as governments, international bodies, relief agencies, non-governmental organizations (NGOs), financial markets and (re)insurance companies, all could benefit from a robust statistical understanding of the co-occurrence of natural hazards. Many also need to manage risks from for the occurrence of damaging events in rapid succession, whose compound impacts may exceed the sum of expected impacts from isolated wet and dry extremes. Similarly, estimates of the range of times that intervene between the two different extremes can inform disaster preparedness and prevention measures. Finally, the growth

of national economies that depend heavily on agricultural outputs and other natural resources such as hydropower can be impacted by sequential hydrological extremes (Zampieri et al., 2017; Zhang et al., 2015).

Notwithstanding their socio-economic relevance, concurrent wet and dry hydrological extreme events at the global scale have seldom been addressed in the literature. One early study did consider combinations of wet and dry extremes via observed PDSI for two thresholds (wet, PDSI > 3 and dry, PDSI < -3) (Dai et al., 2004). This showed that the total global land area (60°S-75°N) impacted by wet-dry extremes increased between 1950 and 2002, with marked changes occurring from the early 1970s and surface warming being identified as the driver of these changes after the mid-1980s. We extend this analysis by: i) using an updated time series (1950-2014); ii) introducing new metrics for assessing concurrent wet-dry extremes; iii) presenting findings at monthly and annual resolution; and above all iv) defining the most geographically-widespread multi-hazard events, occurring within each month, instead of simply considering extreme observations with PDSI > 3 and PDSI < -3. We explore these multi-hazard properties using the monthly self-calibrated PDSI dataset based on the Penman-Monteith model (sc_PDSI_pm) (Dai, 2017; Sheffield et al., 2012). We specifically address the following questions:

i)   To what extent has the global area impacted by wet, dry and concurrent wet-dry hydrological extreme events changed?

ii)  What were the most geographically-widespread extreme wet, dry and concurrent wet-dry events? And what is the associated documentary evidence of extreme conditions during these periods?

iii) How comparatively frequent were wet or dry extremes in the past?

iv)  What is the most likely time interval between opposite extremes at a given location?

v)   How are wet and dry hydrological extremes linked with dominant modes of climate variability?

**2 Data and Methods**

**2.1 Data**

We used the self-calibrated monthly-mean Palmer Drought Severity Index based on the Penman-Monteith model (sc_PDSI_pm) (Dai, 2017; Sheffield et al., 2012) for the 1950-2014 period, at 2.5° horizontal resolution (freely available here). Self-calibration enables a more consistent comparison between different climatic regions, and the Penman-Monteith model outperforms the original PDSI Thornthwaite algorithm (Wells et al., 2004) in representing potential evaporation at the global scale (Sheffield et al., 2012). From this dataset, we obtain extreme wet and dry monthly observed events by conditioning the data on sc_PDSI_pm $\geq$ 3 and sc_PDSI_pm $\leq$ -3, respectively. These two thresholds specify *very moist spells* and *severe droughts*. Only grid-cells with time series having $\geq$ 95% of observations over the period of interest are considered. We acknowledge that the sc_PDSI_pm 2.5° horizontal resolution is relatively coarse, hence highly localised processes, such as convective precipitation in the tropics and mid-latitudes (in summer) may not be well represented. However, we assert that the

sc_PDSI_pm is adequate since our analysis is global and intended to provide a broad overview of concurrent wet and dry extremes.

We further analyse three climate modes of variability known to affect regional and global precipitation patterns: the Niño3.4
(Rayner et al., 2003; Trenberth, 1997), PDO (Mantua and Hare, 2002) and Atlantic Multi-decadal Oscillation (AMO) (Schlesinger and Ramankutty, 1994). All these climate indices are at monthly time-resolution from 1950 to 2014, as issued by the National Oceanic & Atmospheric Administration (NOAA).

## 2.2 Methods for identifying extreme wet, dry, neutral and wet-dry events

First, we calculate the percentage of total land area (km2), derived from our sc_PDSI_pm dataset, impacted by the most widespread monthly extreme wet (sc_PDSI_pm $\geq$ 3) and dry (sc_PDSI_pm $\leq$ -3) hydrological events along with neutral (-3 < sc_PDSI_pm < 3) and extreme wet plus extreme dry events within the period 1950-2014. Monthly extreme wet events were calculated following De Luca et al. (2017) by: (i) computing the wet annual maxima (AMAX), i.e. the highest monthly sc_PDSI_pm observations within each calendar year, at each grid-cell, provided that they satisfy sc_PDSI_pm $\geq$ 3;  (ii)
counting the number of wet AMAX observations occurring on the same date from all the grid-cells (e.g. in December 2010 a total of 217 grid-cells reported a wet AMAX); and (iii) taking the extreme wet event with the most geographically-widespread impacts, i.e. largest impacted area, during 1950-2014. Within all the calculations of the impacted area we considered the difference in area's size of grid-cells at different latitudes. Therefore, we computed the exact grid-cells' area using the *gridarea* function of the Climate Data Operators (CDO) software (freely available here).

Extreme dry events were calculated in a similar way to extreme wet events except that in place of AMAX the sc_PDSI_pm annual minima (AMIN), i.e. the lowest monthly sc_PDSI_pm observations within each year, were used to compute the extreme events, provided that they satisfied sc_PDSI_pm $\leq$ -3. Neutral events were identified as follows: i) extract the sc_PDSI_pm AMAX of (non-extreme) wet events (0 $\leq$ sc_PDSI_pm < 3); ii) extract the sc_PDSI_pm AMIN for (non-
extreme) dry events (-3 < sc_PDSI_pm < 0); iii) pool within the same dataset both (non-extreme) wet/dry AMAX/AMIN events by month; iv) compute the most widespread neutral events by month as per above. Lastly, concurrent extreme wet-dry events were calculated by summing their individual areas for each month. A Mann-Kendall test (Kendall, 1975; Mann, 1945) was performed to assess any significant trends within each time series. Relative Sen's slopes (Sen, 1968) with *p*-values were also computed.

Second, to establish whether the most widespread extreme wet, dry and wet-dry events were solely due to chance, a boot-strapping analysis of n=10,000 samples was performed using the original sc_PDSI_pm dataset. The boot-strapping's steps were as follows: i) prepare the complete global (i.e. all grid-cells) sc_PDSI_pm dataset from 1950 to 2014; ii) sample from all grid-cells together, with replacement, the sc_PDSI_pm values n=10,000 times from this global dataset; iii) calculate n=10,000

wet and n=10,000 dry events with the same algorithm used above for the original dataset; iv) take the impacted area of the most geographically-widespread wet and dry extreme events for each sample; v) calculate the 2.5th and 97.5th percentiles of the extreme wet and dry events' area. Statistical significance was assessed by checking whether observed extreme wet and dry percentage (%) of total impacted areas fell outside the 2.5th and 97.5th percentiles of the boot-strapped events. If this was the case, the observations were considered statistically significant at the 5% level (*p*-value <0.05).

Lastly, to test whether the sc_PDSI_pm values obtained during the most widespread wet, dry and wet-dry events were spatially autocorrelated we computed the Moran's *I* correlation coefficients (Li et al., 2007; Moran, 1950), using the 'geosphere_v1.5-10' and 'ape_v5.3' R packages. The Moran's *I* correlation coefficient can have values between -1 and 1. When $I > 0$ it indicates the existence of clustering between *similar* values; when $I < 0$ it indicates clustering between *dissimilar* values; and when $I = 0$ the values are randomly distributed in space. Statistical significance was assessed under the null hypothesis that there is no spatial autocorrelation between values. Moran's *I* was computed for the most widespread wet, dry, and wet plus dry hydrological extremes.

## 2.3 Wet-dry metrics

The *wet-dry (WD) ratio* is derived on a cell-by-cell basis by taking the natural logarithm of the total number of extreme wet observations (months with sc_PDSI_pm ≥ 3) divided by the total number of extreme dry observations (months with sc_PDSI_pm ≤ -3) over the 1950-2014 period:

$$WD\ ratio_h = \ln\left(\frac{n_{i,h}}{n_{j,h}}\right) \qquad (1)$$

where $h$ refers to a single grid-cell and $n_i$ and $n_j$ are the total number of wet and dry extremes, respectively. The WD-ratio gives information about the propensity of a given area to be more affected by wet or dry extremes. Thus, a WD-ratio > 0 signifies that wet extremes outnumber dry extremes and WD-ratio < 0 indicates a predominance of dry extremes over wet ones. The natural logarithm was used to narrow the range of WD-ratio values and to separate the wet-dominated versus dry-dominated regions by sign. As a caveat, we note that the WD ratio does not account for the different intensities of wet and dry extremes.

*Wet to dry* and *dry to wet* transitions, here named *extreme transitions* (ET) were assessed for each grid-cell by computing the average time interval (months) between these extremes, within the 1950-2014 period. More specifically, ET for wet to dry was derived as follows, for each grid-cell: i) extract both wet (sc_PDSI_pm ≥ 3) and dry (sc_PDSI_pm ≤ -3) extreme observations from the entire (1950-2014) sc_PDSI_pm dataset; ii) order extreme observations by time, from oldest to the most recent; iii) retain only the earliest extreme observation in the case of consecutive extreme dry observations and the latest in the case of

consecutive wet observations; iv) calculate the time interval (monthly difference) between wet and dry extreme observations within the time-series; and v) take the average of the time interval. The same algorithm was applied for calculating ET from dry to wet for each grid-cell, with the only difference being in step iii) where the earliest wet and latest dry extreme observations were kept and in step iv) where the time interval was calculated between dry to wet transitions.

### 2.4 Correlation with Climate Indices

Associations between extreme wet-dry hydrological extremes and the three modes of climate variability (Niño3.4, PDO and AMO) were assessed using the Spearman's rank correlation test (Corder and Foreman, 2014). Specifically, the correlations were performed for each grid-cell only for monthly wet and dry extreme observations (sc_PDSI_pm ≥ 3 and sc_PDSI_pm ≤ -3) within the 1950-2014 period, paired with the corresponding monthly values of Niño3.4, PDO and AMO. Spearman's test does not require data to be normally distributed, making it well suited to the analysis of extreme PDSI values. Since the number of correlation tests performed is large (> 2,700) there is a risk of incurring in statistically significant results simply by chance. Thus, to account for Type I errors (or 'false positives') the Bonferroni correction (Bonferroni, 1936; Sedgwick, 2014) was applied to all *p*-values.

Finally, since Niño3.4 may interact with other modes of climate variability, we removed this signal when correlating the PDO and AMO with sc_PDSI_pm extreme wet and dry observations by performing partial correlations with the R package 'ppcor_v1.1'. Partial correlations represent the relationship between two random variables after removing the effect of one or more other random variables. Here, the partial correlation, between two variables $x_i$ (e.g. PDO) and $x_j$ (e.g. sc_PDSI_pm) given a third variable $x_k$ (e.g. Niño3.4) is defined as follows (Kim, 2015; Whittaker, 2009):

$$r_{ij|k} = \frac{r_{ij} - r_{ik}r_{jk}}{\sqrt{1 - r_{ik}^2}\sqrt{1 - r_{jk}^2}} \qquad (2)$$

Where *r* is the new Spearman's rank partial-correlation coefficient. As a limitation of this approach, we note that our correlations with modes of climate variability do not strictly focus on *concurrent* wet-dry hydrological extremes. Therefore, our results, although in agreement with extreme wet-dry spatial patterns do not explain entirely these multi-hazard events.

### 3. Results

### 3.1 Land area impacted by extreme wet, dry, neutral and wet-dry events

The percentage (%) of total global land area impacted by the most widespread extreme wet, dry and neutral events is shown in Figure 1, at both monthly and annual resolutions from 1950 to 2014. For extreme wet events (sc_PDSI_pm ≥ 3) the average

monthly impacted area over the 65-year period is 2.2% (Figure 1a). The most widespread wet event occurred in December 2010 (7.8%, discussed in Section 3.2). The Mann-Kendall test indicates positive, though non-significant, trends at both monthly and annual scales (Figure 1a-b). The non-significant observed growth in extreme wet area is contrary to previous research, showing a significant decline in (very) wet land areas (Dai, 2011b; Dai et al., 2004). However, varying the

sc_PDSI_pm threshold used to define the extremes, points to the sensitivity of the results to this choice. Indeed, at monthly resolution, when using sc_PDSI_pm $\geq 2$ the wet land area decreases significantly (Sen's slope = -5.4e-04 and $p$-value <0.01, not shown) while, when using a higher threshold of sc_PDSI_pm $\geq 4$ the wet land area increases significantly (Sen's slope = 4.9e-04 and $p$-value << 0.01, not shown).

For extreme dry events (sc_PDSI_pm $\leq$ -3) the average impacted area at monthly resolution is 2.4% and the largest 1-month event occurred in January 2003 (8.6%, Figure 1c and discussed in Section 3.2). In this case, the Mann-Kendall test indicates a positive and statistically significant trend (Sen's slope = 1.7e-03 and $p$-value <<0.01). This signifies that the total area subject to severe drought increased between 1950 and 2014. The trend observed at monthly resolution is stronger, and more evident from the beginning of the 1980s, when aggregating data over annual timescales (Figure 1d, Sen's slope = 2.7e-01 and $p$-value

<<0.01). This result agrees with previous studies showing a global increase in drought risk, attributed to anthropogenic climate change, in both observations and climate model simulations (Dai, 2012, 2011a; Dai et al., 2004; Marvel et al., 2019). Such changes in drought are linked to anomalies in tropical sea surface temperatures (SSTs) and driven by both El Niño and La Niña phases, along with increased surface warming from the 1980s.

The neutral events (-3 < sc_PDSI_pm < 3) affected on average 13.6% of the global land area over the 1950-2014 period, with the largest reaching 30.4% (Figure 1e, monthly resolution). The Mann-Kendall test shows a negative and significant trend (Sen's slope = -1.9e-03, $p$-value <<0.01), once again stronger and more evident at annual timescales (Figure 1f, Sen's slope = -3.2e-01 and $p$-value <<0.01). Such a reduction in the area under neutral conditions is consistent with the observed increasing trend of both extreme wet and dry events. The neutral events show strong seasonality, with peaks of impacted area occurring

during December. The fact that 73.4% of the global sc_PDSI_pm land area is in the northern hemisphere may introduce a bias in the temporal distribution of the extreme and neutral events. For example, boreal and austral winters over the northern and southern hemisphere mid-latitudes are typically wetter than their respective summers. The larger land area in the northern hemisphere means there is a greater chance that more wet events are observed during boreal wintertime (December to February), than during austral wintertime (June to August), thereby driving the peaks in seasonality in Figure 1.

Finally, the area with 1-month concurrent wet-dry hydrological extreme events (Figure 1g) shows an increasing and statistically significant trend (Sen's slope = 1.08e-03 and $p$-value <<0.01), consistent with shorter records (Dai et al., 2004). The mean monthly total global land area with concurrent wet-dry extreme events is 4.6% and the most widespread event impacted a total land area of 13.7% (discussed in Section 3.2). As per dry and neutral events, annually-aggregated data show

a stronger trend (Figure 1h, Sen's slope = 3.0e-01 and $p$-value $\ll$ 0.01) and greater increase in the wet-dry impacted area from the 1980s.

### 3.2 Concurrent global flood and drought events

We next consider the single most extensive wet, wet-dry and dry events, and show that they match reports of severe flood and drought events. The most widespread global extreme wet event was also the most widespread wet-dry event, and occurred in December 2010 (Figure 2a). Recorded events matching this occurrence include the devastating Queensland floods in Australia (BBC, 2010a; Smith et al., 2013; Trenberth and Fasullo, 2012; Zhong et al., 2013), heavy floods and landslides in south-east India which killed more than 180 people (Reliefweb, 2010), widespread flooding and landslides in Colombia and Venezuela causing about 300 deaths and leaving thousands homeless (BBC, 2010b; Telegraph, 2010; Trenberth and Fasullo, 2012) and flooding affecting the north-western USA (NWRFC, 2010). We also find anomalously wet conditions in central-eastern Europe (Figure 2a), although in this region no significant damages were reported by the literature and the media. Such a widespread wet event impacted 7.8% of the total global land area. December 2010 was characterized by a very strong negative Niño3.4 phase, within the 2010-2012 La Niña event (Luo et al., 2017). Moreover, the PDO and AMO were respectively in their cold and positive phases. The same phases occurred during November 2010 (not shown), and these antecedent conditions may have contributed to the extreme wet and dry events in the sc_PDSI_pm series (Lee et al., 2018). At the same time, droughts were recorded in central Asia, Madagascar, the Horn of Africa (BBC, 2011), south America, eastern USA (NOAA, 2011) and north Canada, covering a total of 5.9% of land area. Both the extreme wet and dry percentages (%) of land area impacted (Figure 2a) are significant at the 5% level ($p$-value $<$0.05) according to our bootstrapping procedure (see Section 2.2).

The most widespread extreme dry hydrological event occurred during January 2003 with 8.6% of total land area impacted by drought and 3.8% of land experiencing wet hydrological extremes and floods (Figure 2b). During this event, eastern Australia was the most affected region, with the worst drought in 20 years, driven by an El Niño event that lead to severe dust storms and bushfires (Gabric et al., 2010; Horridge et al., 2005; Levinson and Waple, 2004; McAlpine et al., 2007). This episode belongs to the so called 'Millennium Drought' (Van Dijk et al., 2013) which affected Australia between 2001 and 2009. Other regions experiencing severe drought during January 2003 were north-east China, India (Sinha et al., 2016), Scandinavia (Irannezhad et al., 2017), west Africa, parts of Brazil and a few scattered areas between Mexico, USA, Canada, Russia and Indonesia. January 2003 was an El Niño month with the Niño3.4 index being in a positive phase along with a warm PDO phase. On the other hand, the AMO registered an almost neutral phase. As for the December 2010 episode in Figure 2a, such climate patterns also occurred in the previous month (not shown). Meanwhile, other regions experienced wet hydrological extremes and floods, such as south-east China, central Russia, Europe, southern Great Britain (BBC, 2003; Marsh, 2004), Madagascar (Reliefweb, 2003), Argentina, Chile and scattered parts of Africa and Canada (DFO, 2008). As for Figure 2a, the percentage of land area impacted by both extreme wet and dry events during January 2003 (Figure 2b) was significantly different at the 5% level ($p$-value $<$0.05) from the values expected by chance.

Moran's spatial correlation results are shown in Table 1. Wet and dry extremes in December 2010 (Figure 2a) show $I = 0.24$ (*p*-value <0.001). Positive, statistically significant $I$ results are also obtained when considering wet and dry extremes individually ($I = 0.04$ for wet extremes and $I = 0.25$ for dry extremes, *p*-value <0.001). For the events occurring during January 2003 (Figure 2b) combined wet and dry extremes show $I = 0.27$ (*p*-value <0.001) and wet and dry extremes computed individually have respectively $I = 0.07$ and $I = 0.26$ (*p*-values <0.001). In other words, wet and dry hydrological extremes as both concurrent and independent hazards tend to occur in regions close to each other, with wet and dry, and dry extremes showing greater levels of clustering (or $I$) values (Table 1). This reflects strong spatial coherence between wet and dry extremes, such that a location neighbouring one experiencing drought is more likely to be very dry than very wet (Hannaford et al., 2011).

### 3.3 Wet-dry (WD) ratio

The WD-ratio highlights the 65-year propensity for more or less wet or dry hydrological extremes on a cell-by-cell basis (Figure 3). Hotspots for extreme wet propensity emerge in the USA, northern Mexico, Colombia, Venezuela, Argentina, Bolivia, Paraguay, northern Europe, North Africa, western China, and western and central Australia. On the other hand, regions with higher frequencies of extreme dry events are found in Canada, central south America, central and southern Europe/Africa, eastern China and south-eastern Australia. Other regions, such as Russia, display mixed patterns. These WD-ratio patterns agree with global trends in drought over the period 1950-2010, identified using the sc_PDSI_pm dataset (Dai, 2012).

### 3.4 Extreme transitions (ET)

In Figure 4a we show the average time intervals (months) of extreme transitions (ET) from wet to dry and dry to wet extremes during the period 1950-2014 plotted against the percentage of total global land area affected. The ET from wet to dry (blue curve) exhibits a modal value of 22 months, associated with 4.3% of the total global land area. On the other hand, ET from dry to wet (red curve) peaks at 18-month with ~5% of global land area having this average separation time. Overall, ET from wet to dry takes longer than ET from dry to wet. We also show the cumulative distribution functions (CDFs) of wet to dry and dry to wet ET time intervals (Figure 4b). For wet to dry 50% of the ET occur within ~27 months, whereas for dry to wet half of ET are observed within 21 months. The two ET medians are significantly different (*p*-value <<0.01, two-sided Mann-Whitney-Wilcoxon test) (Mann and Whitney, 1947). The spatial distribution of the natural logarithm (ln) of ET means shows a homogeneous global pattern, without any major regional anomalies (Figure S1). The only noticeable large-scale feature is shorter dry to wet ET means over northern Africa (Figure S1b). Figure 4 also shows few ET > 150 months and these very-long lags between wet (dry) and dry (wet) extremes occurred in different regions across the globe, namely Canada, Brazil, central and southern Africa, India, China and Russia (Figure S2). The reason behind such long ET, that appear to be very localised (i.e. not clustered) events, may be due to precipitation biases in the sc_PDSI_pm dataset or to local geographical characteristics.

ET from dry to dry and wet to wet were also computed (Figure S3). Dry to dry time intervals peak at 27 months with 3.2% of global land area taking this value, whereas wet to wet time intervals peak at 30 months with 3.1% of land area. Half of all dry to dry ET occurred within ~37 months; whereas half of wet to wet ET happen in ~36 months. A two-sided Mann-Whitney-Wilcoxon test shows that the two medians are significantly different (*p*-value <0.01) as per the multi-hazards case.

### 3.5 Correlations with climate indices

In Figure 5, we show global correlations between hydrological extremes (wet and dry) and three of the major modes of climate variability (Niño3.4, PDO and AMO, see Section 2.4). We also computed the same correlation tests for the NAO (Barnston and Livezey, 1987), Pacific North-American (PNA) pattern (Barnston and Livezey, 1987) and Quasi-Biennial Oscillation
(QBO) (Baldwin et al., 2001). However, these results had low statistical significance (Figure S4). Generally, the correlations shown in Figure 5 are consistent with the concurrent wet-dry spatial patterns observed in Figure 2.

ENSO is one of the modes with the most widespread global impacts and is represented here by the Niño3.4 index (Figure 5a). The positive phase of Niño3.4 (associated with El Niño events), is negatively correlated (*p*-value <0.05) with extreme wet
sc_PDSI_pm values over parts of central Canada, northern South America, southern Africa, India, central China, central and northern Russia, Indonesia and eastern Australia. On the other hand, positive significant correlations are found over southern USA, in some isolated regions of central and southern South America and in the Middle East. Such correlation patterns can be explained as a positive Niño3.4 phase associated with wet extremes where correlations are positive and with dry extremes where correlations are negative (this also applies to the positive phase of the PDO and AMO discussed below). The percentage
of total land area impacted by significant Niño3.4 correlations amount to 18.1%. These results agree with the wet and dry patterns linked to ENSO and PDO reported by Wang et al. (2014) for boreal winter (December-February).

Correlations for PDO (Figure 5b, with the Niño3.4 signal removed) partly resemble the spatial patterns found for Niño3.4. Here, negative correlations are also found in north-western North-America, equatorial Africa and eastern Russia, although
most significant correlations over Australia, China and India vanish. Moreover, positive correlations are found in central-western USA, southern South-America and Kazakhstan. The fact that Niño3.4 and PDO correlations show similar spatial patterns (Figures 5a and 5b) suggests that when these two indices are in phase (i.e. El Niño-warm PDO and La Niña-cold PDO), wet and dry extremes are amplified (Wang et al., 2014). The correlation patterns shown in Figure 5b also agrees with season-ahead peak river flow correlations with the PDO (Lee et al., 2018). The PDO significantly impacts a smaller area (12%
of total global land) compared to Niño3.4. Niño3.4 and PDO correlations also tend to resemble the WD-ratio patterns (Figure 3). For instance, we note that both Niño3.4 and PDO show positive rank correlations with the extreme sc_PDSI_pm over the southern and western USA (Figure 5a-b), which are reflected by the predominance of wet extremes (over dry extremes) in Figure 3. Similar patterns are also observed over southeastern Brazil and Argentina. In addition, Figure 5a-b shows negative

correlations with wet extremes over central and eastern Russia, a pattern matched by the predominance of dry extremes (over wet extremes) in Figure 3. Similar coherence in patterns also apply to eastern Australia and central/southern Africa.

The pattern of AMO correlations (Figure 5c, with the Niño3.4 signal removed) differs from Niño3.4 and PDO indices and returns a greater number of significant ($p$-value <0.05) grid-cells (2.5% more overall) and impacted area (18.9%) than Niño3.4. For the AMO, negative and significant correlations are found in Brazil, Argentina, Mexico, scattered areas in north America, the Horn of Africa and eastern China. Positive correlations are found in the Sahel region of Africa, Russia and central Asia. Our results are again in agreement with global, season-ahead correlations reported for peak river flows and the AMO (Lee et al., 2018). We also computed 1-month and 2-month lagged correlations between wet and dry hydrological extremes and Niño3.4, PDO and AMO. The results are qualitatively similar to Figure 5 (Figures S5-S6).

## 4. Discussion and Conclusions

Wet- and dry extremes can coincide in time and/or space creating multi-hazard events that accrue significant socio-economic losses. Geographically remote yet temporally coincident extremes potentially impact stakeholders with global assets and/or supply chains. For instance, knowledge of recurrent patterns of coincident hydrological extremes could be used to hedge losses in regional hydropower production (Ng et al., 2017; Turner et al., 2017) and agricultural yields (Leng and Hall, 2019; Xie et al., 2018; Zampieri et al., 2017) or to manage crop planting dates (Sacks et al., 2010). Rapid successions of extremes at the same location pose challenges for disaster preparedness, event management and long-term risk reduction. Floods and droughts are also expected to become regionally more frequent and severe in the future due to anthropogenic climate change (Arnell and Gosling, 2016; Dai, 2012, 2011a; Hirabayashi et al., 2013; Hirsch and Archfield, 2015; IPCC, 2012; Milly et al., 2002), underscoring the importance of research on concurrent wet-dry hydrological extremes.

We found that the land area affected by extreme dry and geographically remote wet-dry events is increasing with statistically significant trends at both monthly and annual timescales (Figure 1). This matches the expectation that such hazards are likely to increase in the future (Güneralp et al., 2015; Hirabayashi et al., 2013), and is in agreement with previous studies (Dai, 2012; Dai et al., 2004). However, we applied a more stringent definition of extreme events (De Luca et al., 2017) in order to capture well-known flood and drought episodes. We further showed that these extremes can have global-scale impacts, corresponding to documented flooding and drought events, by detecting the most widespread wet, dry and wet-dry events (Figure 2). As a limitation of our study, we recognise that the coarse horizontal resolution of dataset used (sc_PDSI_pm at 2.5°) may not well represent small-scale processes such as localised convective precipitation events.

We introduced two new metrics: the wet-dry (WD) ratio (Figures 3 and S1-S2) and the average time between extreme transitions (ET) for wet/dry and dry/wet extremes (Figure 4). The former reveals the local frequency of extreme wet relative

to extreme dry observations. Areas experiencing more wet than dry extremes were detected in the USA, northern and southern South America, northern Europe and North Africa, western China and most of Australia. More dry than wet extremes were experienced in most of the remaining areas. The ET metric estimates for every grid-cell the average time interval between opposing extremes (i.e. transitions from wet to dry and from dry to wet). The median time between wet to dry transitions is on average slower than the one between dry to wet. Monitoring long-term changes in ET intervals between wet to dry and dry to wet hydrological extremes could provide valuable information on loss accrual and socio-economic impacts.

To this end, it is important to identify possible climate drivers of the observed hydrological extremes. In this study, we computed correlations between wet-dry hydrological extremes and corresponding values of the Niño3.4, PDO and AMO indices (Figures 5 and S5-S6). Our results confirm previous findings about the effect of ENSO, PDO and AMO on global flood hazard and global season-ahead correlations with river peak flows (Emerton et al., 2017; Hodgkins et al., 2017; Lee et al., 2018; Mallakpour and Villarini, 2015; Tootle et al., 2005; Wang et al., 2014; Ward et al., 2014, 2010), while presenting a useful tool for interpreting the most widespread wet-dry events and WD-ratio and ET metrics. PDO spatial correlations with hydrological extremes generally match those of Niño3.4, which support the view that when Niño3.4and PDO are in phase they amplify the global wet and dry changes (Wang et al., 2014). Niño3.4 and PDO correlations also tend to reflect the patterns found for the WD-ratio. In other words, when Niño3.4 and PDO are in a positive/negative phase this leads to extreme wet and dry conditions in some regions and these wet/dry patterns also occur in areas which have in the past experienced respectively more wet/dry conditions. The AMO shows different, and in some cases opposite, correlation patterns when compared to Niño3.4 and the PDO. During the most widespread wet, dry and wet-dry hydrological extremes, the AMO was weak, and indeed the geographical footprint of these events does not closely match that of the AMO. Hence, we assert that the most widespread wet and wet-dry (dry) hydrological extreme events were driven by a La Niña (El Niño) event coupled with a strong negative (positive) PDO phase. However, we note that modes of climate variability cannot explain entirely the physical mechanisms driving these multi-hazard events. Indeed, when selecting similar phase values of Niño3.4, PDO and AMO indices, occurring during the most widespread events (Figure 2), it is not possible to recover similar events in term of overall impacted areas (Figures S7-S8).

The analysis was conducted using the self-calibrated monthly-mean Palmer Drought Severity Index based on the Penman-Monteith model (Dai, 2017; Sheffield et al., 2012). Future research opportunities include using other indices, such as the Standardized Precipitation Index (McKee T.B., Doesken N.J., 1995; McKee et al., 1993) or the Standardized Precipitation Evapotranspiration Index (Vicente-Serrano et al., 2010) to validate our findings and to account for uncertainty in the observations of concurrent wet-dry extremes. Additionally, there is scope to use more recently developed soil moisture metrics emerging from the ESA Soil Moisture CCI Project (Gruber et al., 2019) and NASA Soil Moisture Active Passive (SMAP) mission (https://smap.jpl.nasa.gov/). These datasets have finer spatial-temporal resolution (such as daily at 0.25° for ESA CCI), thus could provide more detailed information about local concurrent wet-dry extremes. Unfortunately, these are not long

enough for trend analysis (e.g. NASA SMAP data begin in 2015). Further work is needed to evaluate the seasonality of the extremes, linked to the modes of climate variability investigated in this work, but also to other ones, such as the Indian Ocean Dipole (IOD) (Saji et al., 1999), which has been recently associated with concurrent floods and bushfires respectively in eastern Africa and Australia (BBC, 2019). Similar analyses could be applied to individual Köppen climate zones (Rubel and Kottek, 2010) to discern possible regional variations in concurrent wet-dry extreme characteristics. Finally, once baseline maps and data for hydrological multi-hazards have been established from observations, the next step should be to use climate model output to evaluate how global patterns of concurrent wet/dry extremes might change in the future.

**Author contribution**

PDL conceived the methods, performed the analyses, created the figures and wrote the first manuscript draft. GM and RW contributed to the methods. All the authors contributed to the writing.

**Competing interest**

The authors declare that they have no conflict of interest.

**Acknowledgements**

PDL was funded by a Natural Environment Research Council studentship awarded through the Central England NERC Training Alliance (CENTA http://www.centa.org.uk/; grant reference NE/L002493/1), Loughborough University and Vrije Universiteit Amsterdam. GM was partly supported by the Swedish research Council Vetenskapsrådet (Grant. No. 2016-03724). The authors would like to thank the two anonymous reviewers and the editor for their constructive comments. PDL would like also to thank Venugopal Thandlam for the useful discussions and comments provided.

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

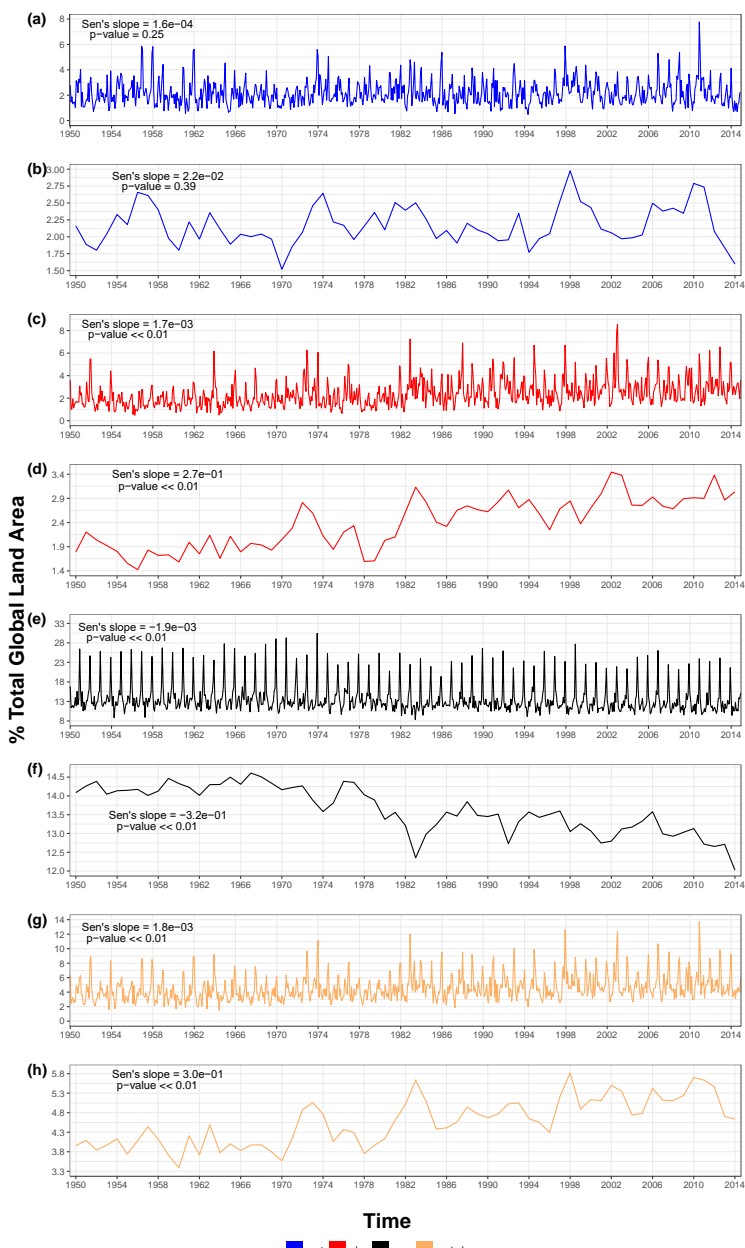

**Figure 1.** Percentage (%) of total land area with **(a)-(b)** wet (blue), **(c)-(d)** dry (red), **(e)-(f)** neutral (black) and **(g)-(h)** extreme wet + extreme dry (orange) events over the 1950-2014 period. Wet extreme events are computed from AMAX having sc_PDSI_pm ≥ 3; dry extreme events from AMIN having sc_PDSI_pm ≤ -3; neutral events from AMAX and AMIN having -3 < sc_PDSI_pm < 3; and wet-dry extreme events by summing the areas of (a)-(b) and (c)-(d). **(a)**, **(c)**, **(e)** and **(g)** show the events at monthly timescale, whereas **(b)**, **(d)**, **(f)** and **(h)** show events aggregated and averaged over annual periods. Sen's slopes and the significance of the Mann Kendall test (*p*-values) are shown in each panel.

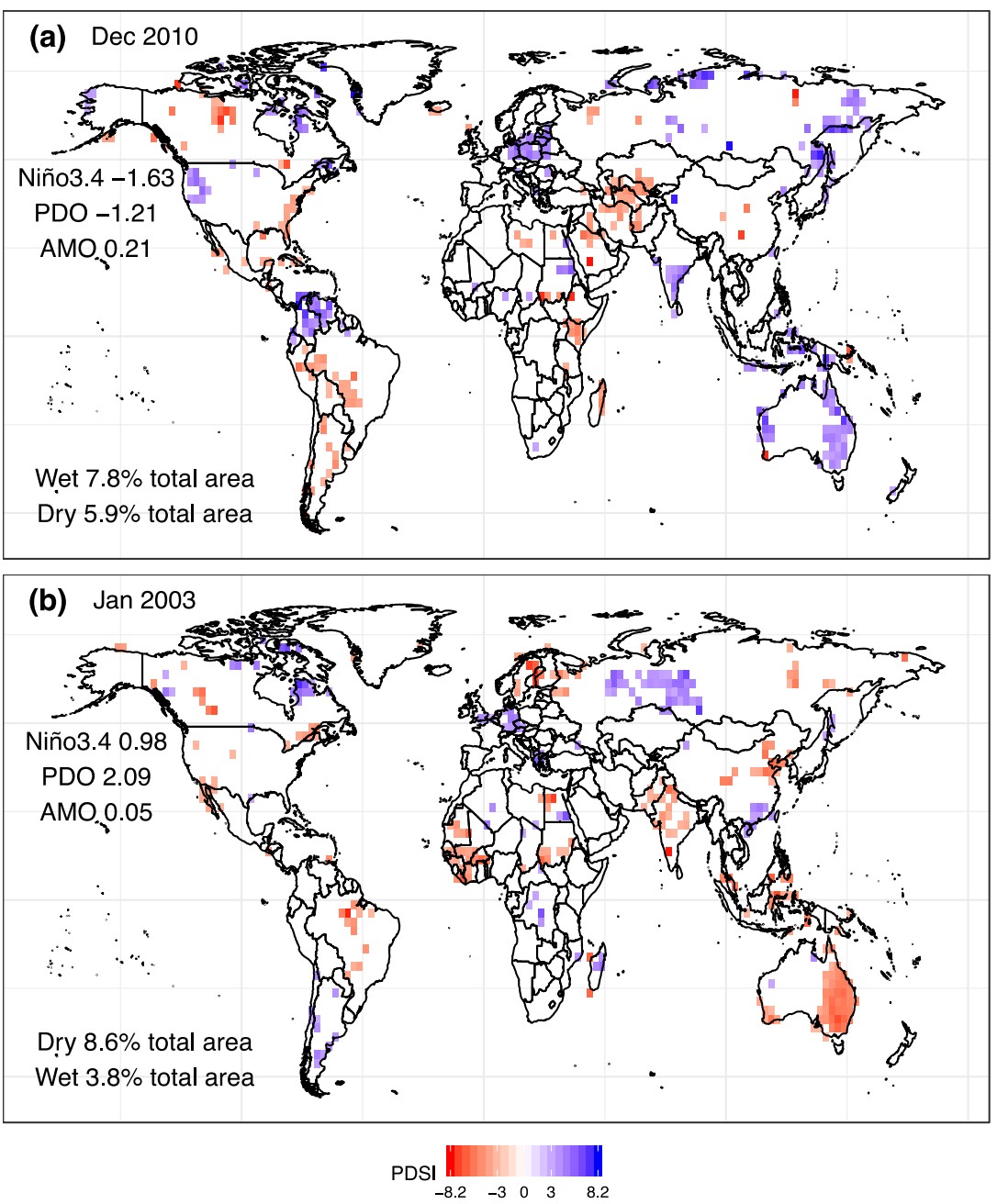

**Figure 2. (a)** Most widespread extreme global wet hydrological event (blue colour) and coincident extreme dry areas (red colour), December 2010. The event was also the most widespread concurrent wet-dry episode. The percentage (%) of total land area is shown for both wet and dry extremes, along with the values of the three climate indices (i.e. Niño3.4, PDO and
5     AMO) in December 2010. **(b)** As **(a)** but for the most widespread extreme global dry hydrological event, January 2003.

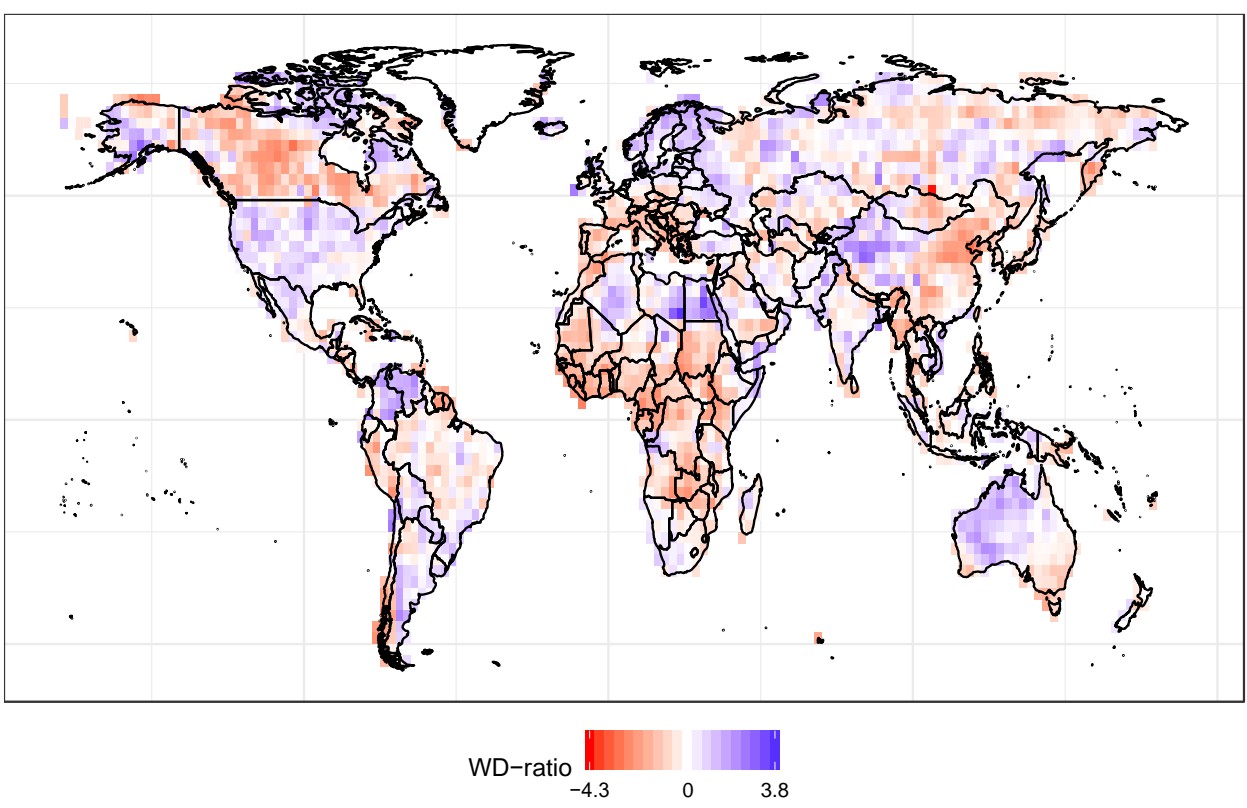

**Figure 3.** Wet-dry (WD) ratio derived for every grid-cell. Blue colours (WD-ratio > 0) mean that the area experienced more wet than dry hydrological extremes. Red colours (WD-ratio < 0) indicate the opposite.

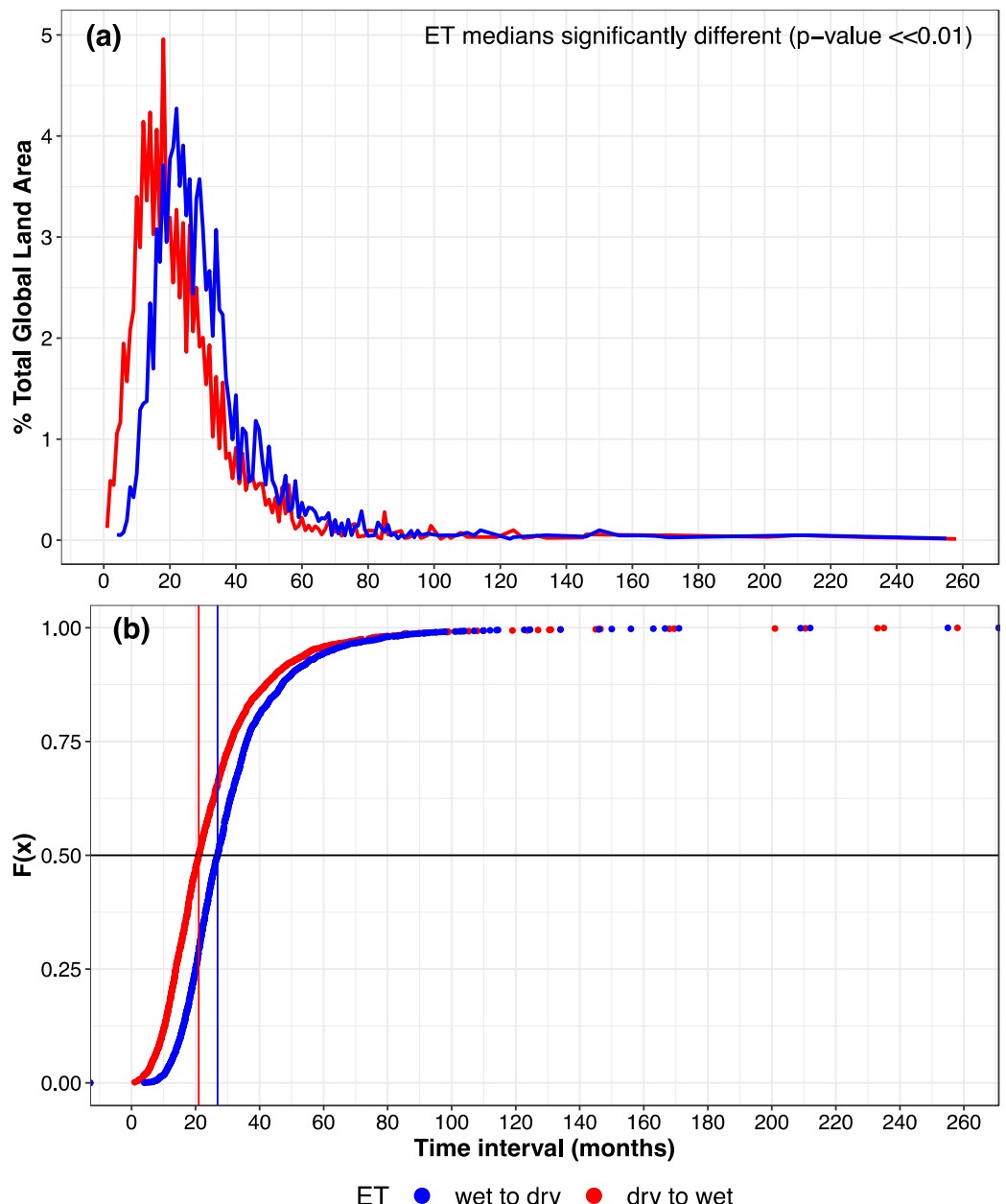

**Figure 4.** Extreme transition (ET) time intervals between extreme wet to dry (blue) and between extreme dry to wet (red). **(a)** Percentage (%) of total land area impacted as a function of ET and **(b)** cumulative distribution functions (CDFs). The vertical blue and red lines mark the medians of the distributions. The two distributions show a statistically significant difference in their medians (*p*-value <<0.01, Mann-Whitney-Wilcoxon test).

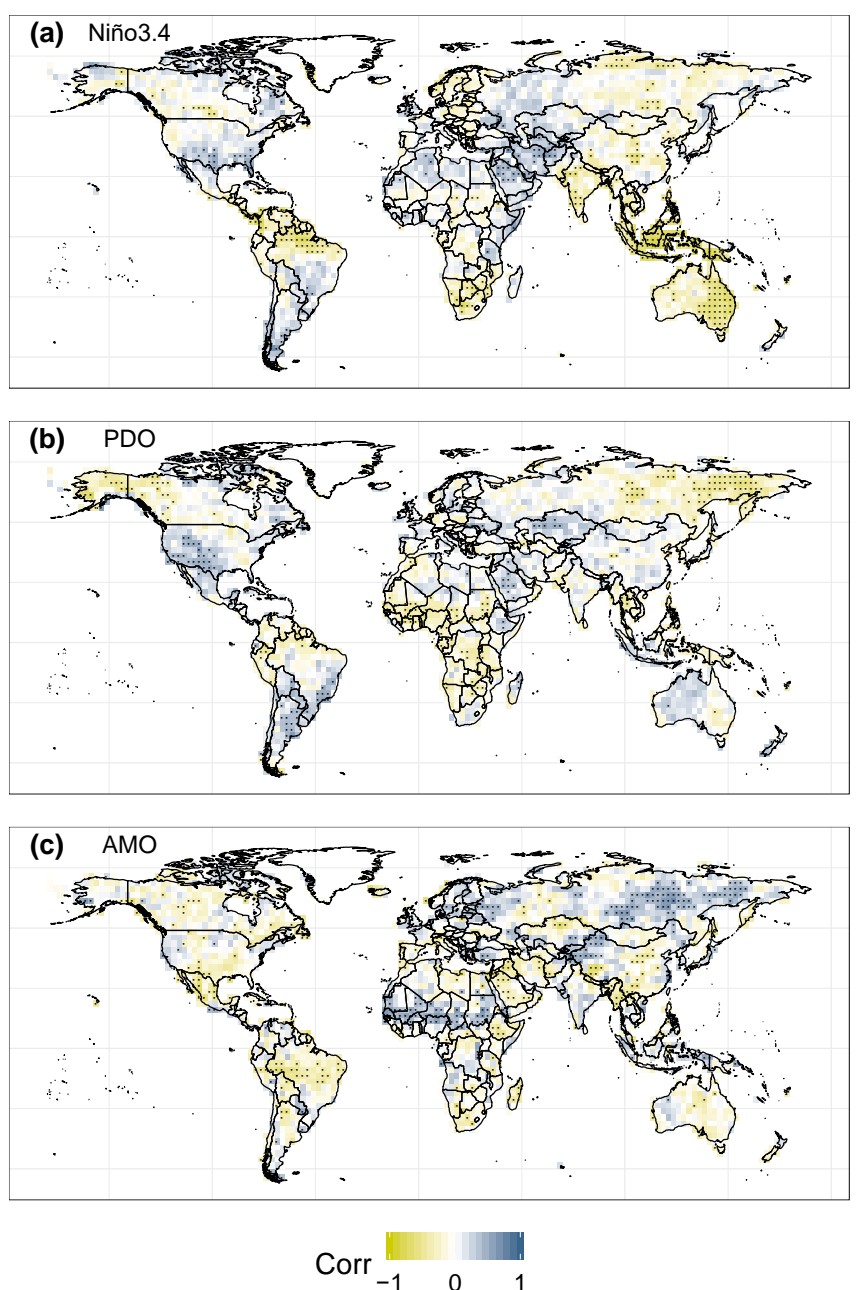

**Figure 5.** Correlations between monthly wet (sc_PDSI_pm ≥ 3) and dry (sc_PDSI_pm ≤ -3) hydrological extremes and **(a)** Niño3.4, **(b)** PDO and **(c)** AMO. For **(b)** and **(c)** partial correlations are performed to remove the Niño3.4 signal. Correlations and partial correlations make use of the Spearman's correlation coefficient. Correlations significant at the 5% level (*p*-value <0.05) are stippled. The Bonferroni correction was applied to all *p*-values.

| Event | Extremes | Moran's $I$ | Standard deviation | $p$-value |
|---|---|---|---|---|
| Dec 2010 | wet and dry | 0.240 | 0.007 | <0.001 |
| | wet | 0.040 | 0.013 | <0.001 |
| | dry | 0.253 | 0.044 | <0.001 |
| Jan 2003 | wet and dry | 0.267 | 0.008 | <0.001 |
| | wet | 0.065 | 0.022 | <0.001 |
| | dry | 0.262 | 0.032 | <0.001 |

**Table 1.** Moran's $I$ correlation coefficients for wet and dry, wet and dry hydrological extremes as in Figure 2a and 2b.