# Peer review of "Concurrent wet and dry hydrological extremes at the global scale"

_Earth System Dynamics, 2019_

## Referee Comment (RC1) · Anonymous Referee #1 · 12 Oct 2019

General Comments

The manuscript is well-organized and contextualized, with an extensive set of citations. The results presented are novel in their comprehensiveness and clearly relevant for a range of societal responses to hydrological extremes, as the authors note. There are various places where additional details are necessary to understand why a particular analytical approach was employed, or to further reflect on the implications of the results. However, these are fairly limited in number, and I am confident that the authors will be able to make appropriate adjustments with relative ease.

Specific Comments

Page 3, line 19: It would seem suitable to mention NGOs as another set of stakeholders

that typically have geographically diverse portfolios. Page 3, line 30: Do the authors mean that surface warming is attributed to changes? Or vice versa? Page 4, line 14: Given that 2.5-deg resolution is relatively coarse, the authors should somewhere have a sentence or two noting why this resolution is sufficient for their survey, or at least listing some regions where it may pose more of a challenge. Page 4, line 31: I found point (ii) unclear – over what time/space ranges is the counting done?

The co-occurrence discussion (e.g. section 3.2) is highly interesting. I wonder, however, if some sense of the closeness/connectedness of the events should be captured in order to truly reflect impacts, which is the motivation that the authors initially present. For example, it is not self-evidently clear why it matters that floods in Australia and the Northwest US, for instance, occurred simultaneously. Relatedly, some sense of the geographic distribution of co-occurring hydrological extremes might be useful in reflecting the 'widespread, simultaneous' character of event that the authors are trying to measure. I was especially struck by Figure 2b, in which it seems that the global peak is largely driven by drought in eastern Australia, while the rest of the world is in fact similar to normal conditions.

In Section 3.5 (correlations with climate indices), the approaches used aren't capable of proving that these modes of variability explain the results. In other words, there may be a wide range of amount of hydrological extremes associated with similar mode-of-variability combinations. Some analysis exploring this issue should be considered.

Page 10, lines 22-23: That the AMO has the largest overall effect is interesting & surprising. What do the authors make of the fact that while the AMO has the largest effect, for the two most extreme wet & dry events (Figure 2) it apparently plays almost no role?

Need for minor methodological comments: The authors should somewhere add thoughts on the usefulness of soil moisture metrics in addition to PDSI, SPEI, etc. Also, how much do they think that their results might be sensitive to the choice of PDSI

threshold? Lastly, calculating lagged correlations with variability modes is probably worthwhile to consider, or at the very least a sentence should be added explaining why this was not done/would not provide much more information.

The Figure 4a caption seems to be inverted. As I would state it, ET is plotted as a function of time interval, not total land area impacted. This figure might also be better posed as comparison against the distribution of times as expected from a random Poisson process. While the comparison between the modes of the wet-to-dry and dry-to-wet distributions is easy enough, for instance, it is not straightforward to interpret what the 'long tail' means – is this tail longer or shorter than would be expected by chance? A Poisson comparison (or some other such reconceptualization) would help in making this figure more intuitive.

---

## Referee Comment (RC2) · Anonymous Referee #2 · 30 Dec 2019

In this study the authors want to identify concurrent wet and dry hydrological extremes at the global scale using PSDI indices from 1950 to 2014. In the study two new metrics are introduced to measure the relative occurrence of extreme wet or dry events and to quantify the time interval between hydrological extremes with opposite sign. The spatial patterns of wet and dry extremes are linked to climate modes, like ENSO, AMO and PDO. The idea of the analysis is interesting and the potential for the results is high, however the manuscript remains mostly descriptive. The fact that the events are concurrent is interesting, but physical explanations should be given. The idea to consider correlation with main climate modes should represent a way to identify possible physical relationship between concurrent events, at least for some regions. In my opinion some revision is needed before the work can be accepted for publication in

the journal. Below detailed comments are listed:

1 - the main weakness of the manuscript is that at the end it results mostly descriptive, lacking some physical explanation for the concurrency of extremes events, at least for some regions; 2 - Fig 2 refers to specific cases (Dec 2010 and Jan 2003). One question is, considering for example the values or phases of PDO, AMO and ENSO, are there are years comparable to 2010? and what happen to the extremes in those years? The same question is valid for the opposite case: are there other years with values of ENSO, PDO and AMO comparable to 2003? and what happen to the extremes in those years? 3 - Fig 2: some regions, like eastern Australia, India, western Africa, Argentina, parts of western US, have opposite (at least in terms of sign) values in the two extreme cases, while others, like central Europe or eastern Canada, have similar values (at least for the sign). Do you have any comments/explanation about that? what about the possible role of large-scale climate modes, in this respect? 4 - How is ET distributed in space? Is there any relationship with the values shown in figures 2 and 3? ET is somehow related to the timescales of the climate modes considered, at least in some specific regions? 5 - Fig. 5: why extreme wet and extreme dry are considered together? Is the signal exactly symmetric in terms of the influence of the climate modes? 6 - Fig. 5: is there any relationship between the regions where the correlation (for each mode considered) is significant and specific behaviors/patterns identified in figures 2 and 3?

Other minor comments: 1 - Lines 27-31: the abstract should contains more specific details about size and shape of the influence of the climate modes considered; 2 - The introduction is apparently too detailed toward the end, but in none line before the definition of the events considered is given; 3 - does the conclusions contain answers to the questions raised from lines 4 to 8? These answers should be clearly highlighted in the Conclusions (and partially also in the Abstract) 4 - Line 14: not clear what kind of product you are using. Is it derived from reanalysis data? I would like to see more details in the description of the dataset used; 5 - Lines 21-22: reference missing or

derived from outputs not shown. Actually it would be interesting to see that; 6 - Line 24: timeserie in fig 1c is largely marked by the seasonal cycle. Visual understanding would be easier considering annual means? 7 - Lines 29-30: meaning not clear. And is this true only for neutral events? 8 - Lines 9-10: should be eastern China and southeastern Australia instead? 9 - Fig 5: last sentence of the caption contains infos already given few lines before in the caption itself

---

## Author Response (AR1)

**Response to the Editor**

We would like to thank the editor and the two anonymous referees for their time and constructive comments and feedbacks, which we believe significantly improved the manuscript. We checked the suggested reference (Boschi & Lucarini, 2019) but unfortunately found no direct links between the flooding events considered by the study and our results. Please, see below the responses to the anonymous referees (1)-(2). As part of the re-submission, we provided a manuscript version with text changes highlighted in red (De_Luca_et_al_track_changes.pdf) and a new Supporting Material file (De_Luca_et_al_Supp_Material.pdf). Both files have been uploaded as a single .zip file (Supp_Material_and_track_changes.zip). In our responses we directly refer to changes in the text using page P and line L abbreviations.

**Responses to Anonymous Referee (1)**

The referee **comments** are highlighted in **black** and numbered with **#1-10**, whereas the **responses** are in **red**.

**General Comments**

The manuscript is well-organized and contextualized, with an extensive set of citations. The results presented are novel in their comprehensiveness and clearly relevant for a range of societal responses to hydrological extremes, as the authors note. There are various places where additional details are necessary to understand why a particular analytical approach was employed, or to further reflect on the implications of the results. However, these are fairly limited in number, and I am confident that the authors will be able to make appropriate adjustments with relative ease.

We thank you for taking the time to revise our manuscript and please, see our responses to your comments below.

**Specific Comments**

**1 Page 3, line 19: It would seem suitable to mention NGOs as another set of stakeholders that typically have geographically diverse portfolios.**

NGOs are now mentioned within the sentence in the revised paper (P3, L29).

**2 Page 3, line 30: Do the authors mean that surface warming is attributed to changes? Or vice versa?**

In the sentence we meant that surface warming was considered as the *cause* (or driver) of the observed wet-dry extremes changes. The sentence has been amended in the revised paper (P4, L8).

**3 Page 4, line 14: Given that 2.5-deg resolution is relatively coarse, the authors should somewhere have a sentence or two noting why this resolution is sufficient for their survey, or at least listing some regions where it may pose more of a challenge.**

Thank you for the comment. The sc_PDSI_pm has indeed a coarse resolution that over certain regions (e.g. the tropics) and periods of the year (e.g. summer in the mid-latitudes) may not well represent for example local convective precipitation events. However, since our study has a global scope, we believe that the sc_PDSI_pm is the dataset that best suits our needs. We mentioned this in Section 2.1 (P4, L32-34 and P5, L1-2) and Section 4 (P12, L29-31) of the revised paper.

**4 Page 4, line 31: I found point (ii) unclear – over what time/space ranges is the counting done?**

The count is computed by summing the number of wet AMAX occurring on the same date (i.e. year-month) from all the grid-cells. For example, in March 1970 a total of 28 grid-cells reported a wet AMAX, in December 2010 a total of 217 grid-cells reported a wet AMAX, etc.. We clarified the sentence in the revised paper (P5, L15-16).

**5 The co-occurrence discussion (e.g. section 3.2) is highly interesting. I wonder, however, if some sense of the closeness/connectedness of the events should be captured in order to truly reflect impacts, which is the motivation that the authors initially present. For example, it is not self-evidently clear why it matters that floods in Australia and the Northwest US, for instance, occurred simultaneously. Relatedly, some sense of the geographic distribution of co-occurring hydrological extremes might be useful in reflecting the 'widespread, simultaneous' character of event that the authors are trying to measure. I was especially struck by Figure 2b, in which it seems that the global peak is largely driven by drought in eastern Australia, while the rest of the world is in fact similar to normal conditions.**

Thank you for your comment. The overall scope of this work is to simply bring to light the possibility that floods and drought can co-occur simultaneously in different regions in the world (see Figure 2). We tried to justify the importance of the impacts of such concurrent extreme events by mentioning the possibility to hedge economic losses with respect to hydropower production, agricultural yields and planting dates (P12, L16-18). However, because of the original motivation of the work, we would like to keep further investigations about concurrent wet-dry extremes *impacts* for later studies. With this work, we indeed hope to have stimulated the interest of a range of academic experts and stakeholders. In Figure 2b it is true that the largest extreme dry area is located in Australia, however there are also other regions, such as India, Indonesia and western Africa, affected by extreme dry conditions. At the same time, we appreciate that at least some exploratory analysis in this direction is appealing.

In the revised manuscript we tested for the *Moran's I* (Moran, 1950; Li et al. 2007) spatial autocorrelation between wet, dry and wet-dry sc_PDSI_pm values in the grid-cells shown in both Figure 2a and 2b. The Moran's *I* correlation coefficient can have values between -1 and

1. When *I* > 0 it indicates the existence of clustering between *similar* values; when *I* < 0 it indicates clustering between *dissimilar* values; and when *I* = 0 the values are randomly distributed in space. We also checked for statistical significance under the null hypothesis that there is 0 spatial autocorrelation between the grid-cells. Generally, all the *I* correlation coefficients are positive and statistically significant (p-values <0.001), meaning that the sc_PDSI_pm values are clustered (or autocorrelated) among similar values. We added these informations in a new Table 1 (P28), Section 2.2 (P6, L7-13), and Section 3.2 (P10, L2-10) of the revised manuscript.

**6 In Section 3.5 (correlations with climate indices), the approaches used aren't capable of proving that these modes of variability explain the results. In other words, there may be a wide range of amount of hydrological extremes associated with similar mode-of variability combinations. Some analysis exploring this issue should be considered.**

For a given state of one or more modes of variability, there could indeed be a wide range of possible hydrological extremes, and we did not want to convey the message that the modes of variability explain the entirety of our results. We further note that the correlations of wet and dry hydrological extremes with modes of climate variability is not strictly connected to the concurrent wet-dry extremes. However, it definitely provides new insights with respect to the global distribution of PDSI extremes associated with a given climate mode. There exists an extensive literature discussing the link between regional hydrological extremes and modes of variability. In the revised study, there are some key references in support of our findings with PDSI (i.e. Wang et al., 2014; Lee et al., 2018) and in Section 2.4 (P7, L23-25), we clarify the limits of our analysis which certainly does not prove causation. We would finally like to note that in our work, three more climate modes have been tested against wet and dry hydrological extremes (i.e. NAO, PNA and QBO), however the vast majority of correlations are not statistically significant. We now show these results in the Supplementary Material (Figure S4).

**7 Page 10, lines 22-23: That the AMO has the largest overall effect is interesting & surprising. What do the authors make of the fact that while the AMO has the largest effect, for the two most extreme wet & dry events (Figure 2) it apparently plays almost no role?**

In Figure 5c the spatial patterns of the AMO influence on wet and dry extremes have been shown along with stippling representing statistical significance (p< 0.05). It is true that in Figure 2, or during the most widespread wet, dry and wet-dry extreme events, the AMO is not in a strong phase (i.e. AMO=0.21 for extreme wet and wet-dry and AMO=0.05 for extreme dry). The disagreement between the large influence of AMO on wet and dry extremes and Figure 2 can be explained by the fact that the grid-cells showing statistically significant correlations with the AMO (Figure 5c) hardly match with the grid-cells reporting extreme wet, dry and wet-dry events (Figure 2). For example, in Figure 2 part of Australia, India, central Europe, northern South-America, the Middle-East and central-western Russia are affected by hydrological extremes and these regions are not the ones showing statistically significant correlations with the AMO (Figure 5c). We therefore conclude that the patterns in Figure 2 are consistent with the fact that the AMO at the time of the two events was weak.

Need for minor methodological comments:

**8 The authors should somewhere add thoughts on the usefulness of soil moisture metrics in addition to PDSI, SPEI, etc. Also, how much do they think that their results might be sensitive to the choice of PDSI threshold?**

We understand the concern raised. To some extent, PDSI, SPI and SPEI can be considered as a proxy for measuring soil moisture, since they are derived from variables such as precipitation, evaporation and temperature. However, they have by no means a perfect correlation with soil moisture. In the paper, we discuss these metrics in the Introduction (P3, L7-22). However, based on your comment, we also discuss in Section 4 of the revised manuscript (P13, L31-34 and P14, L1) soil moisture metrics such as the ones derived from the ESA Soil Moisture CCI Project (Gruber et al., 2019) and the NASA Soil Moisture Active Passive (SMAP) mission (https://smap.jpl.nasa.gov/).

Concerning the second part of your comment, we replicated Figure 1 with two other PDSI thresholds, i.e. 1) PDSI <= -2 & PDSI >= 2 (Figure R1), and 2) PDSI <= -4 & PDSI >= 4 (Figure R2). Results are generally in agreement with the original Figure 1 (PDSI <= -3 & PDSI >= 3), except for the wet observations (Figure 1a) and for the fact that in Figure R1 the area impacted is larger and in Figure R2 is smaller (because more and less observations available). Here, for Figure R1 the wet land area decreases significantly, whereas for Figure R2 it increases significantly. It is worth nothing that in Figure R1 PDSI observations are not extremes, whereas in Figure R2 they are even more extreme than Figure 1. We added two sentences describing this findings in Section 3.1 (P8, L3-7).

[Figure]

*Figure R1 - As Figure 1 in the main manuscript but with wet and dry extreme events defined with a sc_PDSI_pm threshold ≥ 2 and ≤ -2 respectively, and neutral events defined within this range.*

[Figure]

*Figure R2 - As Figure 1 in the main manuscript but with wet and dry extreme events defined with a sc_PDSI_pm threshold ≥ 4 and ≤ -4 respectively, and neutral events defined within this range.*

**9 Lastly, calculating lagged correlations with variability modes is probably worthwhile to consider, or at the very least a sentence should be added explaining why this was not done/would not provide much more information.**

Thank you for the comment. As suggested, we performed lagged correlations with climate modes for 1 and 2 months in advance, i.e. PDSI at t0 and climate modes at t-1 and t-2. Thus, Figure 5 has been replicated with these new lagged correlations (Figures R3-R4). As Figures R3-R4 show, both the correlations and statistical significance patterns are qualitatively similar when compared to Figure 5. We have mentioned this in the revised study

(Section 3.5, P12, L9-10) and added Figures R3-R4 in the Supplementary Material (Figures S5-S6).

[Figure]

*Figure R3 - As Figure 5 but correlations are lagged of 1 month, i.e. PDSI at t0 and climate modes at t-1.*

[Figure]

*Figure R4 - As Figure 5 but correlations are lagged of 2 months, i.e. PDSI at t0 and climate modes at t-2.*

**10 The Figure 4a caption seems to be inverted. As I would state it, ET is plotted as a function of time interval, not total land area impacted. This figure might also be better posed as comparison against the distribution of times as expected from a random Poisson process. While the comparison between the modes of the wet-to-dry and dry-to-wet distributions is easy enough, for instance, it is not straightforward to interpret what the 'long tail' means – is this tail longer or shorter than would be expected by chance? A Poisson comparison (or some other such reconceptualization) would help in making this figure more intuitive.**

The caption of Figure 4a has been amended as suggested in the revised paper (P26, L3). However, we struggled to understand the Reviewer's suggestion to compare our results to a Poisson distribution, and to evaluate whether the tails are longer/shorter than expected by chance. A Poisson distribution is informative in the case of wanting to know the probability of a given number of events occurring in a fixed interval. In our case, this could be the number of land grid-cells with Wet to Dry ET above or below a given threshold. However, the curves in Figure 4 show the % of affected land area for a given ET duration in months. Moreover, a Poisson distribution assumes the events occur independently of previous occurrences, which may often not be true in our case. Concerning comparing the tails of the curves to something happening by chance, this would require some assumptions on the parent distribution of the ET transitions, which may well be grid-cell dependent. What one may do is take a Peak over Threshold (POT) approach and conduct an extreme value analysis of the tails of the distributions, but this would not necessarily answer the Reviewer's point. If we have misunderstood the Reviewer's suggestion, we would be happy to address this point in a second round of revisions.

In this study the authors want to identify concurrent wet and dry hydrological extremes at the global scale using PSDI indices from 1950 to 2014. In the study two new metrics are introduced to measure the relative occurrence of extreme wet or dry events and to quantify the time interval between hydrological extremes with opposite sign. The spatial patterns of wet and dry extremes are linked to climate modes, like ENSO, AMO and PDO. The idea of the analysis is interesting and the potential for the results is high, however the manuscript remains mostly descriptive. The fact that the events are concurrent is interesting, but physical explanations should be given. The idea to consider correlation with main climate modes should represent a way to identify possible physical relationship between concurrent events, at least for some regions. In my opinion some revision is needed before the work can be accepted for publication in the journal. Below detailed comments are listed:

Thank you for taking the time to revise our manuscript. Please, find below our answers to your comments.

**1 the main weakness of the manuscript is that at the end it results mostly descriptive, lacking some physical explanation for the concurrency of extremes events, at least for some regions;**

The main purpose of our work was to bring to light the existence of spatially-remote and concurrent in time wet-dry hydrological extremes. We therefore agree that the manuscript may lack detailed explanation of physical mechanisms driving such multi-hazard events. In the revised manuscript (Section 4, P13, L8-25) we expanded the physical interpretation of our findings, by making use of literature on the impacts of modes of climate variability on regional hydrological extremes.

**2 Fig 2 refers to specific cases (Dec 2010 and Jan 2003). One question is, considering for example the values or phases of PDO, AMO and ENSO, are there are years comparable to 2010? and what happen to the extremes in those years? The same question is valid for the opposite case: are there other years with values of ENSO, PDO and AMO comparable to 2003? and what happen to the extremes in those years?**

We computed new extreme wet-dry maps similar to Figure 2, with ENSO, PDO and AMO values closely matching the ones of Figure 2 (Figures R1-R2). Specifically, we looked for climate modes' values within a +/- 0.3 interval, compared to December 2010 and January 2003, and plotted the corresponding wet and dry hydrological extremes. For example, in Figure R1 we looked for months with -1.33 < ENSO < -1.93, -0.91 < PDO < -1.51 and 0.51 < AMO < -0.09.

There are a total of five months showing similar climate modes' states as for December 2010 (Figure R1) and seven as for January 2003 (Figure R2). Generally, the overall area impacted

by wet and dry extremes is not as high as the one of Figure 2 and the spatial distribution of events differs. This suggests that the extremes highlighted in Figure 2 are not primarily driven by the modes of variability (see also answer to comment #6 by Referee 1). We discussed this in the revised manuscript (Section 4, P13, L22-25) and added Figures R1-R2 in the Supplementary Material (Figures S7-S8). We hope that this answers your question, but we would be happy to investigate this topic further in a second review round following additional comments that may arise.

[Figure]

*Figure R1 - Wet and dry hydrological extremes occurring during similar (+/- 0.3) climate modes of variability phases as per the ones of Figure 2a (December 2010).*

[Figure]

*Figure R2 - Wet and dry hydrological extremes occurring during similar (+/- 0.3) climate modes of variability phases as per the ones of Figure 2b (January 2003).*

**3 Fig 2: some regions, like eastern Australia, India, western Africa, Argentina, parts of western US, have opposite (at least in terms of sign) values in the two extreme cases, while others, like central Europe or eastern Canada, have similar values (at least for the sign). Do you have any comments/explanation about that? what about the possible role of large-scale climate modes, in this respect?**

The fact that same regions in Figure 2a show extremes of opposite sign (i.e. wet and dry) compared to Figure 2b is totally plausible, since every area (or simply grid-cell) is neither *always* experiencing wet nor dry conditions. Climate modes of variability can indeed provide an explanation to this and by looking at both Figure 2 and Figure 5 we note that their patterns are in agreement with the most widespread wet, dry and wet-dry events.

For example, in Figure 5a-b we note that eastern Australia and eastern Asia show significant negative correlations between the positive phases of ENSO and PDO, and wet extremes. This pattern is mirrored in Figure 2a, where these areas experience wet extremes during the negative phase of ENSO and PDO. Similarly, in the middle-East, positive ENSO and PDO phases are significantly and positively correlated with wet extremes (Figure 5a-b) and in Figure 2a, during *negative* ENSO and PDO phases, the area is experiencing extreme dry conditions. The same concepts apply for example to India and northern South America (Figure 2a and Figure 5) and generally also between Figure 2b and Figure 5. We added a sentence in the revised manuscript (Section 3.5, P11, L10-11) highlighting the agreement between Figure 2 and Figure 5. At the same time, in view of our reply to comment #2 above, one should not overstate the role of the climate modes of variability. Indeed, we do not recover the patterns shown in Figure 2 by simply selecting months with similar combinations of variability indices.

**4 How is ET distributed in space? Is there any relationship with the values shown in figures 2 and 3? ET is somehow related to the timescales of the climate modes considered, at least in some specific regions?**

We computed maps showing the spatial distribution of ET, i.e. Wet to Dry and Dry to Wet (Figure R3). Figure R3 shows the natural logarithm (ln) of ET means. We made use of the ln because ET data has a large positive skewness, thus the ln transformation would make the interpretation of the maps easier. A qualitative comparison between Figure R3 and Figures 2-3 does not show any particular agreement, however we would like to keep Figure R3 in the manuscript and thus it has been added in the Supplementary Material (Figure S1), along with two sentences describing it in the revised paper (Section 3.4, P10, L28-30). We also added in the Supplementary Material (Figure S2) the equivalent of Figure R3 but with ET computed without transformation to ln, to highlight the most extreme ET (> 150 months, Figure R4). We also discussed this in Section 3.4 (P10, L30-33).

ENSO shows interannual variability, whereas PDO and AMO are characterised by multidecadal variability. The ET medians are ~27 months for wet to dry and 21 months for dry to wet. Thus, there is no immediate link between the ET and the timescales of modes of climate variability. This, naturally, does not exclude some forced periodicity in ET resulting from the influence of the modes of variability, but we reserve a systematic statistical analysis of this for a future study.

[Figure]

*Figure R3 - Maps for (a) wet to dry and (b) dry to wet extreme transitions (ET). The colours show the natural logarithm (ln) of ET means for each grid-cell.*

[Figure]

[Figure]

*Figure R4 - As Figure R3 but with ET not transformed with natural logarithm (ln).*

**#5** Fig. 5: why extreme wet and extreme dry are considered together? Is the signal exactly symmetric in terms of the influence of the climate modes?

In Figure 5 the correlations between hydrological extremes and modes of climate variability have been computed, for each single grid-cell, by correlating time-series of both extreme wet and extreme dry observations (all together) with the time-series of the given climate mode. We computed the correlations in this way because the extreme wet and dry time-series are, on average for each grid-cell, symmetric, with 46.8% of extreme wet and 53.2% of extreme dry observations. By having such symmetry between wet and dry extremes one can compare the observations with the time-series of modes of climate variability, which also show a symmetry between positive and negative values by definition.

**6 Fig. 5: is there any relationship between the regions where the correlation (for each mode considered) is significant and specific behaviors/patterns identified in figures 2 and 3?**

Yes, there is a plausible link between the significant correlation patterns (Figure 5) and the most widespread wet, dry and wet-dry hydrological extremes (Figure 2). Please see the answer to your comment #3 above.

On the other hand, linking Figure 5 with the WD-ratio (Figure 3) is not trivial since the two figures show different processes. However, we can note that for ENSO and PDO, positive correlations with wet extremes are observed over the southern and western USA (Figure 5a-b), a pattern which is somehow reflected by the predominance of wet extremes (over dry extremes) in Figure 3. Similar patterns are also observed over southeastern Brazil and Argentina. In addition, Figure 5a-b shows negative correlations with wet extremes over central and eastern Russia, a pattern matched by the predominance of dry extremes (over wet extremes) in Figure 3. Similar coherence in patterns also seems to apply to eastern Australia and central/southern Africa.

Thus, one can genuinely speculate on the fact that ENSO and PDO correlations are in agreement with the WD-ratio patterns. In simpler words, when ENSO and PDO are in a positive/negative phase this leads to extreme wet and dry conditions in some areas around the globe and these wet/dry patterns also occur in areas which in the past experienced respectively more wet/dry conditions. We added these observations and discussion in the revised manuscript in Sections 3.5 (P11, L30-33 and P12, L1-2) and Section 4 (P13, L15-18).

Other minor comments:

**7 Lines 27-31: the abstract should contains more specific details about size and shape of the influence of the climate modes considered;**

In the revised manuscript we added to the abstract (P1, L32-33 and P2, L1) and Section 3.5 (P11, L20,29 and P12 L5) the percentage of the statistically significant areas impacted by the climate modes and in the abstract we also listed the most impacted areas (P2, L1-3).

**8 The introduction is apparently too detailed toward the end, but in none line before the definition of the events considered is given;**

We dedicated a short subparagraph in the Introduction mentioning the definition of our wet, dry and wet-dry events and why they may be important for impact studies (P3, L24-33 and P4, L1-2). The *event* definition has been clarified within the text (P3, L24-28).

**9 does the conclusions contain answers to the questions raised from lines 4 to 8? These answers should be clearly highlighted in the Conclusions (and partially also in the Abstract)**

Thank you for your comment. We revised the research questions (P4, L15-21) and in the new manuscript version the answers to the questions are clearly stated, following the original sequence (i-v), in the Abstract and in Section 4.

**10 Line 14: not clear what kind of product you are using. Is it derived from reanalysis data? I would like to see more details in the description of the dataset used;**

There are several versions of the Palmer Drought Severity Index (PDSI). In our work we used the self-calibrated PDSI based on the Penman-Monteith model. The publication linked to the dataset is the following and we now updated it correctly in the revised paper: *Dai, A., 2017. Dai Global Palmer Drought Severity Index (PDSI). Research Data Archive at the National Center for Atmospheric Research, Computational and Information Systems Laboratory. Accessed 23/04/2019. https://doi.org/10.5065/D6QF8R93*. We also added the web-link for accessing the dataset in the revised paper (P4, L27). Moreover, we discuss in detail the dataset and now mention its limitations also based on comment #3 of Referee 1 (Section 2.1 P4, L32-34 and P5, L1-2 and Section 4 P12, L29-31).

**11 Lines 21-22: reference missing or derived from outputs not shown. Actually it would be interesting to see that;**

The reference for the Mann-Whitney-Wilcoxon test has been added to the revised manuscript (P10, L28). However, since comment 5 has no page specified if you were referring to a different reference please do not hesitate to let us know and we will amend the text accordingly.

**12 Line 24: timeserie in fig 1c is largely marked by the seasonal cycle. Visual understanding would be easier considering annual means?**

It is true that Fig 1c shows a marked seasonal cycle and certainly aggregating the data over annual means would make the interpretation easier, as the overall trend is stronger. However, since Fig 1c shows neutral PDSI observations, or -3 < PDSI < 3 which by definition are not considered extremes (and therefore they are less impactful), we would like to keep Figure 1c (now Figure 1e) as it is, also for consistency with the other panels (Figure 1a-b,d, now Figure 1a,c,e,g). The choice of showing monthly instead of annual observations, has been done on purpose: i) to match the PDSI time-series; ii) to show the single largest event for each month; and iii) to provide as much observations as possible to the reader. However, since when considering data aggregated over annual timescales the seasonal cycle is not present and it also shows a clear decline/increase in neutral/dry and wet-dry land area impacted since the 1980s, we added the annual trends in Figure 1 (see Figure R5) and discuss the new findings in the revised manuscript (Section 3.1).

[Figure]

*Figure R5 - New Figure 1 with percentage (%) of total global land area (y-axis) aggregated over annual time-scale shown in (b)-(d)-(f)-(h).*

**13 Lines 29-30: meaning not clear. And is this true only for neutral events?**

Most of the global land area is located in the northern hemisphere. Therefore, there is higher chance that neutral and/or extreme events are observed over this hemisphere. For example, during boreal/austral winters the weather is known to be particularly wet over the northern/southern hemispheres. Thus, since in the northern hemisphere there is more land (and therefore more grid-cells from where to obtain PDSI time-series) there is higher chance that most of the extreme wet events are recorded in the northern hemisphere. Such abundance of extreme wet events in the northern hemisphere introduces an asymmetry in the temporal distribution, or seasonality, of the events. We clarified with more details this concept in the text of the revised paper by expanding the original sentence and by adding an example (P8, L24-28).

**14 Lines 9-10: should be eastern China and southeastern Australia instead?**

Yes, thank you for spotting this. The sentence has been amended in the revised paper (P10, L17).

**15 Fig 5: last sentence of the caption contains infos already given few lines before in the caption itself.**

The sentence has been removed. Thank you.

[revised manuscript text omitted]

---

## Author Response (AR2)

**Response to the Editor**

Thank you very much for accepting our manuscript for publication in ESD. We mentioned and added the Indian Ocean Dipole (IOD) reference as suggested in Section 4 of the manuscript.